# Environment Curriculum Generation
# via Large Language Models

**William Liang, Sam Wang, Hung-Ju Wang,**
**Osbert Bastani, Dinesh Jayaraman[†], Yecheng Jason Ma[†]**
GRASP Laboratory, University of Pennsylvania
wjhliang@seas.upenn.edu

https://eureka-research.github.io/eurekaverse/

**Abstract:** Recent work has demonstrated that a promising strategy for teaching robots a wide range of complex skills is by training them on a curriculum of progressively more challenging environments. However, developing an effective curriculum of environment distributions currently requires significant expertise, which must be repeated for every new domain. Our key insight is that environments are often naturally represented as code. Thus, we probe whether effective environment curriculum design can be achieved and automated via code generation by large language models (LLM). In this paper, we introduce `Eurekaverse`, an unsupervised environment design algorithm that uses LLMs to sample progressively more challenging, diverse, and learnable environments for skill training. We validate `Eurekaverse`'s effectiveness in the domain of quadrupedal parkour learning, in which a quadruped robot must traverse through a variety of obstacle courses. The automatic curriculum designed by `Eurekaverse` enables gradual learning of complex parkour skills in simulation and can successfully transfer to the real-world, outperforming manual training courses designed by humans.

**Keywords:** LLMs, Curriculum Learning, Environment Design, Quadrupeds, Sim-To-Real RL

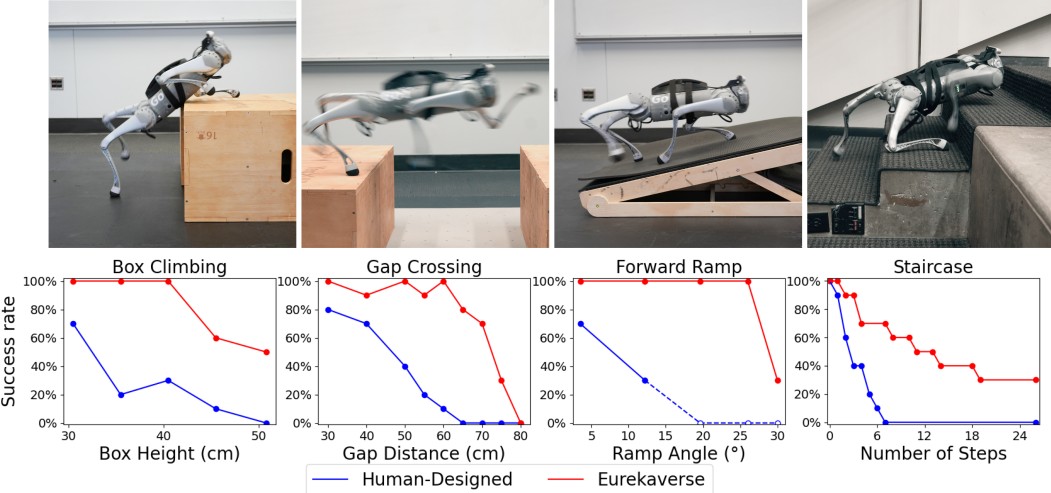

Figure 1: Policies trained on `Eurekaverse` parkour courses significantly outperform those trained on `Human-Designed` across four distinct real-world tasks, each with a variety of difficulties.

---

[†]Equal Advising

8th Conference on Robot Learning (CoRL 2024), Munich, Germany.

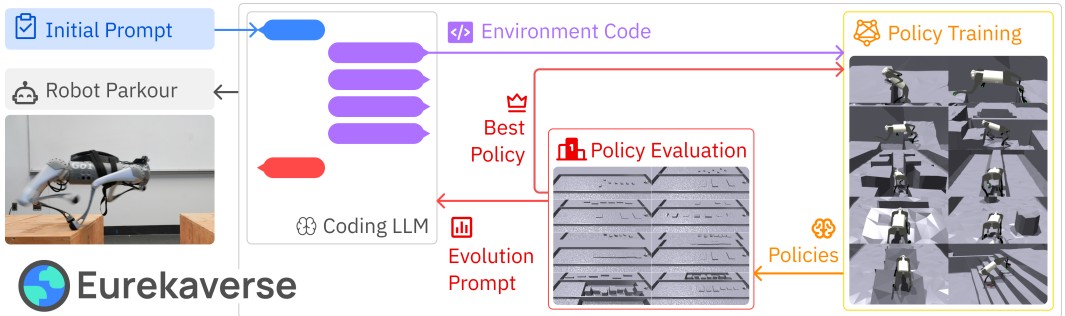

Figure 2: `Eurekaverse` automatically learns complex skills by performing agent-environment co-evolution, which iterates between evolutionary environment generation and population-based policy training and evaluation.

## 1 Introduction

It is often hypothesized that the evolution of intelligent motor control in biology is driven by the need to habituate in and adapt to varied environment conditions [1, 2]. Inspired by this observation, training robot policies over a curriculum of environments has been shown to be an effective strategy for the acquisition of complex robot skills [3–7]. Designing a useful curriculum of environments is, however, time-consuming and requires domain-specific knowledge to work well [8]. In limited cases, when the environment variations can be succinctly represented by scalar parameters [9, 10], unsupervised environment design methods have shown capability in automatically generating progressively more challenging tasks in low-dimensional simulation domains [9, 11, 12]. However, whether these methods can scale up to challenging real-world robotic tasks is an open question.

Our key insight is that, in many practical robotics scenarios, environment variations can be represented by diverse programs. This allows us to solve environment design using large language models (LLMs) since they have been shown to excel at code generation in diverse domains [13–16]. We introduce `Eurekaverse`, a language model powered environment curriculum design algorithm that can generate and evolve environments in code space. At a technical level, `Eurekaverse` instructs an LLM to generate novel environments to teach the target task, trains policies for each environment, and evaluates the best-performing policies to inform the LLM when it generates the next round of environments. Given that large language models (LLMs) have exhibited strong code generation and self-improving capabilities on other domains [17–19], we hypothesize that using LLMs for environment curriculum design could also enable learning complex robot skills. In particular, we validate `Eurekaverse`'s efficacy on the task of quadrupedal parkour [20, 21], in which a quadruped robot is tasked with traversing diverse and unknown challenging courses with varied terrains and obstacles such as gaps, hurdles, boxes, and ramps. In order to train a policy deployable in the real world, prior parkour methods carefully designed training courses in simulation that enable real-world policy transfer [20, 21]. Because parkour courses are represented as programs that specify detailed course terrain and geometry (see Fig. 3), we posit that parkour is an ideal testbed for testing `Eurekaverse`'s capability in handling the complexity of evolving environment programs. Parkour involves a rich variety of challenging locomotion skills so our results could also be of independent interest for scaling up locomotion learning.

We demonstrate that the automatically generated curriculum of environments from `Eurekaverse` leads to policy training that continuously improves over time without plateauing; in contrast, baselines that only train over a fixed set of terrains, designed by either humans or an LLM, tend to overfit on these training terrains, resulting in worse generalization to test simulation and real-world courses. Specifically, on a set of 20 held-out simulation test courses that are carefully designed to holistically evaluate robot parkour skills, `Eurekaverse`-trained policies outperforms various baselines and ablations. Furthermore, on several real-world courses, `Eurekaverse`-trained policies successfully transfer and exhibit more robust and adaptive behavior than policies trained using limited or manually designed training courses.

In summary, our contributions are:

1. `Eurekaverse`, an LLM-based unsupervised environment design algorithm that can automatically generate curriculums of environment programs for robot skill learning.

2. Extensive simulation and real-world validation of `Eurekaverse` on quadrupedal robot parkour.

## 2  Related Work

**Large Language Models for Robotics.** Recent work has demonstrated that LLMs can be used in robotics domains in various ways. They have demonstrated capabilities as high-level semantic planners [22–34] and control policies [35–37]. Leveraging the code generation capability of LLMs, recent work has also explored using LLMs to guide policy learning via reward function design [38–41] and environment design [42–44]. In this latter category, prior work shows that LLMs can generate varied environments for top-down pick-and-place tasks that enable semantic generalization [42] or diverse tasks across different embodiments without clear intra-task dependencies [43]. In contrast, our work is the first to demonstrate how LLMs can be used in an evolutionary approach to generate full environment *curricula* that can guide the learning of complex physical skills such as parkour. Finally, Ma et al. [44] proposes using LLMs to automatically sample physics parameter randomization for sim-to-real policy learning [45–47]; physics randomization is a restricted class of environment design, involving selecting a value range for each physics parameter in an enumerated list. Our domain specification here is much larger, more complex, structured: it is a full-fledged program. In our specific case study of robot parkour, this program specifies one out of a very large space of environment geometries.

**Environment Design and Curriculum Learning.** Beyond using LLMs for environment program generation, environment design and curriculum learning have been extensively studied in prior literature. Prior works have considered framing environment design as a multi-agent game between the environment generator and the policy [9–12, 48–51]; however, these methods require manually designed environment generators and can handle only a small space of environment variations (e.g., layouts in a 2D maze), limiting their scalability to challenging problems such as parkour that has a large and complex design space. Curriculum learning has been used in robotics to learn complex skills [5, 52–56], but prior works use manually designed curricula or confined environment variation space that require extensive domain expertise and tuning. Our work demonstrates how to scale up adaptive environment curriculum for challenging physical skills by using LLMs to automate environment generation and modification.

## 3  Problem Setting

In this section, we formalize the unsupervised environment design setting as introduced in [9]. First, we model a fully specified environment as a Partially Observable Markov Decision Process (POMDP). Here, an environment is a tuple $M = (A, S, R, O, \mathcal{T}, \mathcal{I}, \gamma)$, where $A$ is the space of actions, $S$ is the set of states, $O$ is the set of observations, $R : S \times A \to \mathbb{R}$ is the reward function, $\mathcal{T} : S \times A \to \Delta(S)$ is the transition dynamics function, $\mathcal{I} : S \to O$ is the emission function, and $\gamma$ is the discount factor. Given $M$, the goal of reinforcement learning is to learn a policy $\pi : S \to \Delta(A)$ that maximizes the cumulative discounted sum of rewards: $V^M(\pi) = \mathbb{E}_{\pi, \mathcal{T}}[\sum_{t=0}^{T} \gamma^t R(s_t, a_t)]$.

Then, a POMDP *template* is a tuple $\mathcal{M} = (A, S, R, O, \Theta, \mathcal{T}, \mathcal{I}, \gamma)$, where $\Theta$ represents the space of environment variations that can be incorporated into the transition dynamics $\mathcal{T} : S \times A \times \Theta \to \Delta(S)$. While environment configuration can include parameter variations such as simulation physics values [44], in this work, we focus on semantic and geometric variations (e.g., different parkour terrains) that are represented as programs. Given this, an environment *curriculum* can be parameterized by a sequence of programs $\vec{\theta} = (\theta_1, ..., \theta_T) \in \Theta^T$. Then, in unsupervised environment design (UED), we are interested in an *environment generator* $\Lambda : \Pi \to \Theta$ that can generate a curriculum of environments $\vec{\theta}$ for the policy $\pi \in \Pi$ to continue learning. The goal of UED is to

```
You are a reinforcement learning engineer trying
to write environment functions for a quadruped
robot parkour task...            System Prompt

def set_terrain(length, width, difficulty, ...):
  height_field = np.zeros((length, width))
  goals = np.zeros((8, 2))
  plat_length = 1.0 - 0.3 * difficulty
  ...
  def add_platform(...):
  ...
  for i in range(6):  # Set up 6 platforms
    dx, dy = np.random.randint(...), ...
    add_platform(...)
    # Put goal in the center of the platform
    goals[i+1] = [
      cur_x + (platform_length + dx) / 2,
      mid_y + dy
    ]
  ...
  return height_field, goals      In-context Example
```

```
def set_terrain(length, width, difficulty, ...):
  platform_length = 1.0 - 0.2 * difficulty
  platform_width = np.random.uniform(1.2, 1.6)
  ...
  def add_platform(start_x, end_x, mid_y, height):
    ...
  def add_ramp(start_x, end_x, mid_y, ...):
    ...
  for i in range(6):
    platform_height = np.random.uniform(...)
    add_platform(cur_x, cur_x + ...)
    if i % 2 == 0: add_ramp(cur_x, cur_x + ...)
    else: cur_x += gap_length
  ...                              LLM Output
```

Figure 3: Our prompt and in-context example (blue), an example LLM response (purple), and its visualization. In the rendering, large red dots indicate goals, and the blue dot is the current goal; small dots indicate heading command (direction to the goal).

learn a policy that can *zero-shot* generalize to unseen environments, maximizing $\sum_i V^{M_i}(\pi)$ for a set of unseen environments $\{M_{\text{test},i}\}$ parameterized by $\Theta_{\text{test}} = \{\theta_{\text{test},i}\}$. Therefore, $\Lambda$ must be able to generate both learnable and useful environments for policy learning in order to maximize generalization capability.

In this work, we investigate whether LLMs can be an effective choice of $\Lambda$ for designing robot parkour courses for quadrupedal robots, where $\Theta$ can be thought of as the space of programs that captures different terrain configurations.

## 4 Methods

At a high level, `Eurekaverse` proceeds as follows. First, it uses the LLM to generate an initial set of environments to train on. Then, it uses a process we call *agent-environment co-evolution*, where it iteratively uses reinforcement learning (RL) to train agents on the current set of environments, followed by updating the set of environments so they continue to challenge the best current policy without being too difficult for learning. Intuitively, as the best current policy learns to act more effectively, the environments are evolved so they can more effectively improve this policy. This strategy resembles a traditional curriculum learning pipeline, with the key difference being that the curriculum itself is generated automatically by the LLM.

Throughout this section, we ground the exposition in the context of quadruped parkour to concretize the method discussion; we note that the general algorithmic principle is broadly applicable to other environment design problems. Implementation details and pseudocode are in the Appendix.

### 4.1 Initial Environment Generation

In order to design an effective environment curriculum, we require a mechanism for generating an initial set of diverse environments. We first query the LLM, given a description of the task and a single in-context example environment program; then, the LLM responds with an environment program $\theta$. To obtain a diverse set of training environments, we can sample the LLM multiple times with nonzero temperature. In particular, given in-context example $\theta_{\text{incontext}}$, we sample environments

$$\theta_j \sim \Lambda_{\text{LLM}}^{\text{init}}(\theta_{\text{incontext}}), \quad j = 1, ..., NJ,$$

where $N$ is the number of RL agents and $J$ is the environments per agent (see Sec. 4.2 for details).

In quadruped parkour, this configuration $\theta_j$ is an obstacle course terrain, specified as a height field—a 2D grid $H$ that defines the height of coordinates along the ground plane—and a list of goal coordinates $C = (x_1, y_1), \ldots (x_8, y_8)$. Both $H$ and $C$ are implemented as Python code and defined via for loops and Numpy operations. Due to this programmatic format, our generation procedure also includes a simple validity check for maximum height and maximum height difference thresholds to filter out obviously impossible or incorrect terrains; details are in Appendix. See Figure 3 for the generation prompt, in-context example, LLM output, and environment rendering.

## 4.2 Co-Evolution of Agents and Environments

The initial environments can already be used for training RL agents. However, a key issue is that the LLM has zero knowledge about the RL agent's capabilities and limitations, so they may not be very useful for training. Furthermore, they are sampled i.i.d., so they are unlikely to form a progressively more challenging curriculum. Indeed, as we show in our experiments, a single policy trained on these environments is sub-optimal, as much of the training budget is likely spent on environments that are too difficult or too easy for the agent's current capabilities.

To address this issue, we propose a *co-evolution* approach of environments and agents. Specifically, this process alternates between training RL agents to improve on the current set of environments, and using the LLM to evolve the set of environments to further improve the agent. This alternating strategy naturally leads the environments to form a curriculum for the corresponding RL agents. We describe both of these steps in more detail below.

**Policy evolution.** At the beginning of each iteration, we use RL to train a population of $N$ agents $\{\pi^i\}_{i=1}^N$, each on its own training library of $J$ environments $\{\theta_j^i\}_{j=1}^J$; this training happens independently and in parallel. Then, we evaluate each agent on the union of all environments $\Theta_{\text{proxy}} = \cup_i \{\theta_j^i\}_{j=1}^J$ across all iterations, and select the best-performing policy $\pi^{\text{best}}$. In the first iteration, the RL agents are randomly initialized; in the subsequent iterations, they are initialized to $\pi^{\text{best}}$ from the previous iteration. Thus, the policies progressively become more effective across iterations.

Intuitively, maintaining a population of agents increases the likelihood that some agents are trained on useful environments and not affected by bad environments; then, evaluating policies on the union of all environments selects the one that generalizes best to unseen environments.

**Environment evolution.** In the first iteration, the initial environments generated as described in Sec. 4.1 are randomly split among the $N$ RL agents. In subsequent iterations, we create new environments by evolving those that were used to train $\pi^{\text{best}}$, $\{\theta_j^{\text{best}}\}_{j=1}^J$. Since $\pi^{\text{best}}$ is effective, these environments are known to be effective for training. We then use the LLM to evolve them: for each $\theta_j$ in this set, we provide the LLM $\theta_j$ and ask it to produce a variation on that environment:

$$\tilde{\theta}_j \sim \Lambda_{\text{LLM}}^{\text{evol}} \left( \theta_j, \theta_{\text{incontext}} \right). \tag{1}$$

Here, $\theta_j$ serves as a previous LLM response, and our LLM prompt contains additional information including environment statistics (e.g. maximum terrain height difference) and policy training statistics (e.g. reward terms and success rate) as well as an in-context evolution example. See Appendix for prompt and details. We perform this procedure independently $N$ times, which produces a new set of training libraries $\{\theta_j^i\}_{j=1}^J, i = 1, \ldots, N$ for the $N$ policy training runs in the following iteration.

# 5 Experiments

## 5.1 Experimental Setup

**Simulation Training.** We adopt the simulation framework from Cheng et al. [21], which trains parkour policies for the Unitree Go1 robot on obstacle course terrains defined by a height field and goal coordinates. During training, terrains are instantiated at multiple difficulties affecting obstacle dimensions and spacing. Policies are initialized at low difficulties, and they are promoted or demoted to higher or lower difficulties depending on the number of goals reached. This procedure gradually

introduces the policy to harder obstacles and ensures that the thousands of parallel training robots cover a wide, diverse set of courses to avoid overfitting behaviors at local minima.

Using PPO [57], we train a teacher policy that takes in privileged scandot sensing (i.e., terrain heights at specific positions around the robot). To deploy, we distill the teacher into a student that receives depth frames from a front-facing camera. All methods and ablations use the same reward and training setup. Additional training details are in the Appendix.

**Algorithm Details.** `Eurekaverse` uses GPT-4o as the LLM, and we run 5 iterations of generation, each with 8 parallel policy training runs of 2000 steps. To teach our policy a stable walking gait (which requires reward terms not relevant to parkour, e.g., torso orientation), we first pre-train a simple walking policy with 1000 policy update steps. One full run of our method takes around 24 hours on 8 A6000 GPUs, each with 48 GB VRAM, and incurs an OpenAI API cost of around $15.

**Methods.** We compare `Eurekaverse` with the `Human-Designed` environments from Cheng et al. [21], which were also designed for a generalist quadruped parkour agent. We emphasize that this comparison does not take into account the time and training steps required to first design `Human-Designed` optimally, whereas `Eurekaverse` must perform both environment design and policy training within the allocated steps. Additionally, we compare to a `Random` baseline that places randomly positioned ramps and boxes; these obstacles are sized randomly between half and double the bounding box of the quadruped, making them feasible for a well-trained policy. Finally, we consider an `Oracle` policy trained directly on the testing environment. `Random` serves as a lower bound—confirming whether environment design is actually necessary—and `Oracle` serves as an upper bound—if we have access to the true testing environment and can directly train on it.

**Ablations.** We also compare against various ablations to probe the importance of our algorithm components. First, we run a variant that always asks the LLM to make the environments more challenging, without providing policy performance feedback (**No Feedback**). Next, we consider training a policy for the same number of steps on only the initial set of LLM-generated environments (**Initial Envs**), the set of environments in the final iteration (**Final Envs**), and a set of environments where each terrain is generated sequentially, conditioned on previously-generated terrains, to maximize diversity (**Diversity Only**). Finally, we train a policy on the in-context example provided to the LLM (**LLM Example**).

## 5.2 Simulated Parkour Benchmark

To compare the zero-shot generalization performance of different policies in simulation, we design a parkour benchmark suite consisting of 20 diverse parkour tasks as our testing environments. These tasks are variations of real world obstacles seen in prior robot parkour learning works, such as box climbing, forward ramps, sideways ramps, A-frames, box jumps, stepping stones, staircases, narrow passages, agility poles, and balance beam [21, 58–63]. Note that these benchmark environments are independent from the `Human-Designed` ones and not revealed to the LLM for environment generation, holistically testing all methods' generalization capability. For each of the 20 tasks, we instantiate 10 versions scaled by difficulty. The benchmark is visualized in the Appendix.

## 5.3 Simulation Experiments Results

In Figure 4 (left), we report the benchmark performance of `Eurekaverse`, `Human-Designed`, and `Random` across training steps as well as `Oracle`'s best performance. In Figure 4 (right), we compare our method's final performance with that of various ablations. Additionally, in Figure 5, we compare `Eurekaverse`'s iterations with `Human-Designed` on each benchmark obstacle.

**Environment design is necessary for parkour.** First, we see that the `Random` baseline does not increase performance on the benchmark, indicating it fails to learn in randomly-generated environments; thus, careful environment design is needed to learn complex quadruped parkour skills.

**`Eurekaverse` outperforms human-designed environments.** On average, our method's final policy achieves nearly 2 additional goals over the policy trained on human-designed environments.

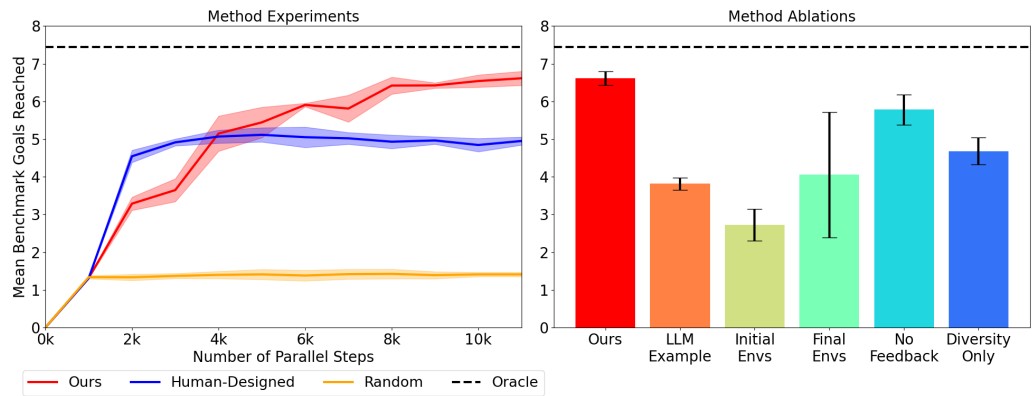

Figure 4: Comparing sim benchmark performance across training steps for `Eurekaverse` and baselines (left), and final benchmark performance for `Eurekaverse` and ablations (right). The training curve for ablations is in Appendix. Experiments are run over 3 seeds.

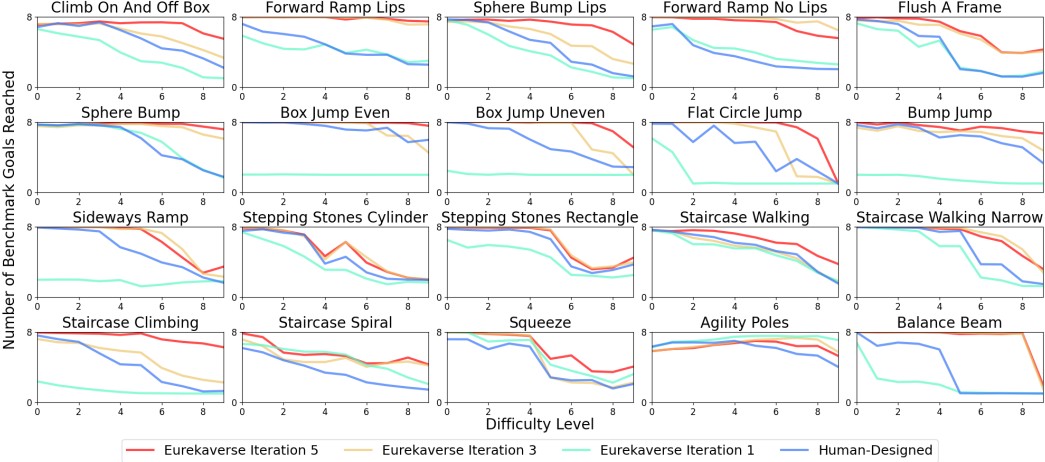

Figure 5: Comparison of `Eurekaverse`'s iterations against `Human-Designed`, visualized per obstacle type and over each difficulty (easiest to hardest) on sim benchmark. Higher area under the curve is better.

The latter learns quickly but plateaus; intuitively, though well-designed, the relatively small number of distinct terrains limits the capacity for generalization. On the other hand, while our policy learns slower in the first few iterations due to imperfect initial environments, it exhibits a stronger upward trend since the environments are iteratively adapted to the current best policy, allowing the policy to continuously improve. Similarly, we see in Figure 4 (right) that `Eurekaverse` surpasses **LLM Example**, the human-created in-context example for the LLM queries; while the well-designed example teaches the policy to succeed in a specific subset of benchmark tasks, the LLM can use it to creatively construct a more diverse array of environments, thereby teaching a more generalist and performant policy.

**`Eurekaverse` nearly matches the oracle.** After 5 iterations of design, our policy nears the performance of `Oracle` in Figure 4, despite not receiving any information about the benchmark.

**`Eurekaverse` generates a curriculum for continuous learning.** From Figure 4 (right), we see that **Initial Envs** and **Final Envs** achieve subpar results. Thus, neither teaches the policy a complete set of parkour skills; the former was generated without information on the policy's training progress, and the latter is tuned for a policy that has already learned skills in previous iterations, making it unsuitable for a beginner policy. Additionally, **Diversity Only** underperforms, showing that diverse environments alone, without an adaptive curriculum, are insufficient for optimal performance. **No Feedback** is also inferior; though it improves similarly to `Eurekaverse` in the initial iterations, policy performance degrades as the evolved terrains fail to match the policy's capabilities (see Appendix). Thus, a curriculum tailored to policy training is necessary, rather than one that simply

increases difficulty. By building a curriculum that bridges the initial and final environments while increasing diversity and taking into account policy feedback, `Eurekaverse` trains a policy that gradually improves over each iteration; we see evidence of this in Figure 5, where our policy generally improves at each benchmark obstacle type.

### 5.4 Real-World Experiments

**Robot Deployment Details.** We deploy on the Unitree Go1, a quadrupedal robot with 12 degrees of freedom. Note that the Go1 weighs 12kg and is equipped with motors that have a maximum torque of 23.70 Nm, which is significantly lower than prior work [20, 21] with the A1, whose motors reach 33.50 Nm and also weighs 12kg. Additional details are in Appendix.

**Real-World Tasks.** Figure 1 shows `Eurekaverse`'s and `Human-Designed`'s performance on box climbing, gap crossing, forward ramp, and stairs tasks, four representative tasks in our simulation benchmark as well as prior works [20, 21, 59, 60]. Box climbing involves getting on top of a box of varying heights, and gap crossing involves jumping from one platform to another. In stairs, the robot must traverse as many steps of stairs as possible, and in forward ramp, the robot must step onto and go across a ramp, with difficulty determining the ramp angle.

**Results.** We deploy distilled policies from `Eurekaverse` and `Human-Designed` and compare their success rates across the four tasks, each of which vary in difficulty corresponding to the key obstacle property. In box climbing, gap crossing, and forward ramp, we perform 10 trials for each difficulty; in stairs, we perform 10 trials total and measure the number of steps traversed. For each obstacle, we initialize the robot at 5 preset locations with varying distance to the obstacle.

In all tasks, `Eurekaverse` generally outperforms `Human-Designed` across difficulties, succeeding at jumps up to 75cm (above the Go1 length), climbing up over 50cm (above the Go1 height), walking up a 30 degree ramp, and traversing the entire staircase. Our policy is also much more safe and stable than `Human-Designed`, which often trips our controller's motor power protection fault; in the forward ramp task, `Human-Designed` incurred physical damages on our robot hardware, so we refrained from completing the remaining trials (dotted line in Figure 1).

We also observe `Eurekaverse` exhibiting recovery behavior, where the policy quickly reacts to slipping or misplaced feet; for example, if it misses a jump, it quickly kicks up its hind legs, allowing the front feet to grab a more stable footing and climb up. Moreover, in the staircase trials, the policy is robust to seeing chairs and railings on either side, despite never seeing similar features in simulation. We hypothesize that this behavior is the result of two mechanisms: during training, policies fail but have a chance to recover before the environment resets, and during depth distillation, random blackout and noise augmentation increases robustness against out-of-distribution depth readings.

## 6  Conclusion

We have presented `Eurekaverse`, a novel method for automated environment curriculum design with large language models (LLMs). `Eurekaverse` generates effective environment programs that adapts to the policy's current training progress, enabling it to learn new skills and hone existing ones. We validate our method on quadruped parkour, where our policy outperforms prior human-designed ones in both zero-shot simulation and real environments. We believe that `Eurekaverse` demonstrates the potential for LLMs to generate an infinite variation of environments, enabling continuous policy learning and serving as a path toward truly open-ended, generalist robot agents.

**Limitations.** Our system requires a moderate amount of LLM samples to consistently generate and select valuable environments; a important future direction necessary for scaling up is increasing sample efficiency, for example by fine-tuning the LLM. Additionally, our evolution prompt only uses textual feedback. A potential extension can additionally provide environment visualizations, leveraging recent multimodal foundation models to potentially perform stronger spatial reasoning.

## 6.1 Acknowledgements

This project was supported in part by NSF CAREER Award 2239301, NSF Award 2331783, and ONR award N00014-22-12677. We would like to thank Ge Yang, Gabriel Margolis, Alan Yu, and Yajvan Ravan for their insights and help in setting up and tuning sim-to-real transfer and deployment infrastructure. We would also like to thank Unitree Robotics (Wen Shuo and Tony Yang) for their support in repairing our robot.

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

# A Appendix

## A.1 Algorithm Details

We provide additional implementation details for `Eurekaverse` below. We first describe the core algorithm loop in pseudocode. Next, we describe "soft" policy selection, a more robust way to select the best policy using evaluation on the proxy (all generated training) environments.

---

**Algorithm 1** Eurekaverse

---
1: **Require**: RL algorithm $\mathcal{A}$, coding LLM `LLM`, performance criteria $F$, LP transformation $G$
2: // Generate initial environments
3: $\theta^i_j \sim \Lambda^{\text{init}}_{\text{LLM}}(\theta_{\text{incontext}})$
4: // Run iterations of co-evolution
5: **for** generation iteration $t = 1$ to $T$ **do**
6:      // Train multiple policies on environments
7:      **for** run $i = 1$ to $M$ **do**
8:          $\pi_i = \mathcal{A}(\{\theta^i_j\}^J_{j=1})$
9:      // Construct evaluation environment and select best policy
10:      $\Theta_{\text{proxy}} = \cup_i \{\theta^i_j\}^J_{j=1}$ across iterations
11:      $\pi_{\text{best}} = \arg\max_{\pi_i} F(\pi_i, \Theta_{\text{proxy}})$
12:      // Evolve environments
13:      $\tilde{\theta}^{\text{best}}_j \sim \Lambda^{\text{evol}}_{\text{LLM}}(\theta^{\text{best}}_j)$
14: **Output**: best final policy $\pi^{\text{best}}$

---

**Soft Selection.** During co-evolution, we select the best policy $\pi^{\text{best}}$ via performance on all generated environments $\Theta_{\text{proxy}}$, which tests the policies' generalization and serves as a proxy for the true testing environment. Empirically, we find that this proxy accurately orders the policies, where the best policy in $\Theta_{\text{proxy}}$ also performs the best in our simulation benchmark. Nonetheless, there are rare cases where the best-performing benchmark policy is not the best one chosen by the proxy. To increase robustness against these inaccuracies, we "softly" select the best policy during each iteration of co-evolution: the selected policy is chosen randomly, where the best policy has probability $p_1$, second-best has probability $p_2 < p_1$, and so on. Each agent in the next iteration performs this selection independently (so that $Np_1$ are initialized to the best policy, $Np_2$ are initialized to the second-best, etc.). In practice, we select $p_1 = 0.75, p_2 = 0.25, p_3, \ldots, p_N = 0$.

## A.2 Prompts and Examples

```
You are a reinforcement learning engineer trying to write environment functions as effectively
    as possible for a quadruped robot parkour task. Please keep in mind that this robot will
    be trained in simulation and deployed in a real world obstacle course, so we want the
    obstacles to be realistic and challenging.

To do so, you should perform the following steps:
1. Carefully read the problem statement, specifications, and tips below.
2. Explain what your obstacle course will look like. Keep in mind that your obstacle sizes
    should be relative to the quadruped's size, and the entire obstacle should fit within the
    terrain bounds. You should be creative when designing your obstacles, and You may draw
    inspiration from features seen in dog parks, playgrounds, and urban environments.
3. Write a function that generates the obstacle course according to your plan.

Problem Statement:
You are given a quadruped robot that needs to navigate through an obstacle course in 3D space.
    The floor of the course is represented as a 2D numpy array, height_field, where each
    element represents the height of the ground at that point, in meters. To direct the
    quadruped, the course has 8 goals that the quadruped must reach in order. The goal
    locations should be stored in a 2D numpy array, goals, as (x, y) indices in height_field.

To create your course, you will write a function set_terrain(length, width, field_resolution,
    difficulty) that creates and returns the height_field and goals arrays. The function
    takes in the length and width of the course, in meters, the scale of quantization for the
    obstacle course, in meters, and the difficulty of the obstacle course, a float between 0
    (easiest) and 1 (hardest) inclusive. Please follow the template below to write your
    function:
'''python
```

```
import numpy as np
import random

def set_terrain(length, width, field_resolution, difficulty):
    """Description of your course and the tested skill."""

    def m_to_idx(m):
        """Converts meters to quantized indices."""
        return np.round(m / field_resolution).astype(np.int16) if not (isinstance(m, list) or
     isinstance(m, tuple)) else [round(i / field_resolution) for i in m]

    height_field = np.zeros((m_to_idx(length), m_to_idx(width)))
    goals = np.zeros((8, 2))

    # Your code here

    return height_field, goals
```

You can use any function from the numpy and random libraries as well as any in-built Python
    functions. Please write everything as Python code and annotate your code with comments,
    including a one-line docstring after the function definition that summarizes your
    obstacle course.

Environment Specifications:
1. height_field quantizes the ground plane into a grid of points. field_resolution = 0.05, in
    meters, is the quantization of the ground's (x, y) axes, which represents the distance
    between adjacent points in the height_field array.
2. The ground plane size is 12 x 4 meters, so the height_field.shape = (12 / field_resolution,
    4 / field_resolution) = (240, 80). Please make sure that your obstacle course spans
    exactly these dimensions.
3. The quadruped's standing size (length, width, height) is 0.645 x 0.28 x 0.4 meters. Keep
    these dimensions in mind when designing the size of your obstacles.
4. The quadruped will spawn with its center at (x, y, z) = (1, width / 2, 0) meters. You must
    place obstacles at indices with x >= 2 / field_resolution to avoid the quadruped spawning
    inside an obstacle.
5. Please make sure that your obstacles have a width of at least 1 meter. However, in rare
    cases, we will allow narrow obstacles with length and width of at least 0.4 meters and no
    smaller, as long as the quadruped is not expected to climb up or down.
6. The goal coordinates should be within the bounds of the course. Even if you have fewer than
    8 obstacles, you must set all 8 goal positions in the goals array.
7. The quadruped will be instructed to travel in a straight line from one goal to the next.
    Thus, if you want the quadruped to turn, you must place a goal at the turning point.
8. Your course should test a particular one of the robot's skills, for example by repeating a
    single type of obstacle. Make the course relatively consistent throughout. There will be
    other courses to test other skills and different obstacles.
9. The heights in field_height can be negative. The quadruped's spawning area will always be
    at a height of 0 meters, so you can use negative heights to create pits or gaps in the
    terrain. Use this to force the quadruped to walk or jump across the tops of your
    obstacles without climbing up or down.
10. Since each index in the terrain has one height value, it is impossible to create overhangs
    or tunnels. Do not try to create these features in your course.

Tips:
1. Do not call your function. Only write the function definition and the code inside it.
2. To broadcast a 1D array to a 2D array (or 2D slice), you must first add a new axis to the 1
    D array.
3. You should not add noise directly to height_field. Random noisy terrain is not necessary
    for our task, and we do not want it as an obstacle.
4. When slicing into height_field, make sure to convert from meters to quantized units
    beforehand. For example, a 2 x 2 meter slice looks like [x - m_to_idx(1):x + m_to_idx(1),
    y - m_to_idx(1):y + m_to_idx(1)].
5. If you write helper functions, please make them nested functions of set_terrain. Your
    response should be a self-contained function definition following our template.

Listing 1: System Prompt. In the function template, we provide a conversion function m_to_idx to simplify the conversion from coordinates, in meters, to the index in the array.

```
The following is an example of a terrain generation function. Please reference the example
    provided, but make your terrain different.

```
import numpy as np

def set_terrain(length, width, field_resolution, difficulty):
    """Multiple platforms traversing a pit for the robot to climb on and jump across."""

    def m_to_idx(m):
        """Converts meters to quantized indices."""
```
```

```
        return np.round(m / field_resolution).astype(np.int16) if not (isinstance(m, list) or
     isinstance(m, tuple)) else [round(i / field_resolution) for i in m]

    height_field = np.zeros((m_to_idx(length), m_to_idx(width)))
    goals = np.zeros((8, 2))

    # Set up platform dimensions
    # We make the platform height near 0 at minimum difficulty so the quadruped can learn to
     climb up
    platform_length = 1.0 - 0.3 * difficulty
    platform_length = m_to_idx(platform_length)
    platform_width = np.random.uniform(1.0, 1.6)
    platform_width = m_to_idx(platform_width)
    platform_height_min, platform_height_max = 0.0 + 0.2 * difficulty, 0.05 + 0.25 *
     difficulty
    gap_length = 0.1 + 0.7 * difficulty
    gap_length = m_to_idx(gap_length)

    mid_y = m_to_idx(width) // 2

    def add_platform(start_x, end_x, mid_y):
        half_width = platform_width // 2
        x1, x2 = start_x, end_x
        y1, y2 = mid_y - half_width, mid_y + half_width
        platform_height = np.random.uniform(platform_height_min, platform_height_max)
        height_field[x1:x2, y1:y2] = platform_height

    dx_min, dx_max = -0.1, 0.1
    dx_min, dx_max = m_to_idx(dx_min), m_to_idx(dx_max)
    dy_min, dy_max = -0.4, 0.4
    dy_min, dy_max = m_to_idx(dy_min), m_to_idx(dy_max)

    # Set spawn area to flat ground
    spawn_length = m_to_idx(2)
    height_field[0:spawn_length, :] = 0
    # Put first goal at spawn
    goals[0] = [spawn_length - m_to_idx(0.5), mid_y]

    # Set remaining area to be a pit
    # We do this to force the robot to jump from platform to platform
    # Otherwise, the robot can just jump down and climb back up
    height_field[spawn_length:, :] = -1.0

    cur_x = spawn_length
    for i in range(6):  # Set up 6 platforms
        dx = np.random.randint(dx_min, dx_max)
        dy = np.random.randint(dy_min, dy_max)
        add_platform(cur_x, cur_x + platform_length + dx, mid_y + dy)

        # Put goal in the center of the platform
        goals[i+1] = [cur_x + (platform_length + dx) / 2, mid_y + dy]

        # Add gap
        cur_x += platform_length + dx + gap_length

    # Add final goal behind the last platform, fill in the remaining gap
    goals[-1] = [cur_x + m_to_idx(0.5), mid_y]
    height_field[cur_x:, :] = 0

    return height_field, goals
```

Listing 2: Initial Example. This example demonstrates the main structure of a terrain function, including helper functions and a for loop over multiple obstacles.

```
We trained a quadruped policy to perform parkour on the obstacle course created by the
    generation code above (as well as others), and we tracked the values of individual reward
     components as well as other metrics such as the number of goals reached (out of 8),
    episode length, and the rate of edge violations (feet too close to edge of obstacles):
<INSERT POLICY STATISTICS HERE>

We have also computed statistics for the terrain height of the direct path between goals
    across multiple difficulties. Note that this is a heuristical shortest-path between goals
     that approximates the locations an optimal quadruped would traverse. The statistics do
    not reflect the actual path taken by the quadruped policy, nor does it include the height
     of gaps, pits, and other obstacles that the quadruped would not step on:
<INSERT TERRAIN STATISTICS HERE>
```

```
Please carefully analyze the statistics above and provide a new, improved generation function.
     You should pay attention to which parts of the course the quadruped successfully learned
     and which parts it struggled with. The goal of your course is to balance difficulty and
     feasibility for the quadruped robot, allowing it to learn and perform better. Thus, if
     the robot is getting stuck on a certain obstacle or goal, you should consider changing or
     removing it. You should also follow the guidelines below:
- If the number of reached goals is over 80%, please create a harder course while ensuring
     that it is feasible and safe for a real robot. You should consider adding more obstacles,
      increasing the complexity of the course, and increasing the difficulty of existing
     obstacles. For example, you can make climbing obstacles taller or jumping gaps wider.
- If the number of reached goals is below 20%, please create an easier course by decreasing
     the difficulty of existing obstacles or simplifying the course layout. For example, you
     can make climbing obstacles shorter or jumping gaps narrower. Please also double-check
     that the course obstacles are fair and feasible for the quadruped.
- Otherwise, please create a variation of the current course with the same difficulty but
     different obstacles. If the quadruped seems stuck on a certain obstacle, please change it
     .

Again, please be creative when designing your course as we want to provide a diverse set of
     training environments for the quadruped. Here is a list of the courses and skills that
     the quadruped was already trained on. You may use them as inspiration, but please make
     sure yours is different:
<INSERT TERRAIN DESCRIPTIONS HERE>

Please use the same template for the course generation function and provide a detailed
     reasoning of the changes you made. The function signature should remain the same.
```

Listing 3: Evolution Prompt. Here, we insert policy training statistics including reward term values, episode length, number of goals reached, and number of edge violations (feet too close to edge) both before and after training. We also compute height field (terrain) statistics, including maximum value, maximum difference between consecutive indices, and standard deviation. Finally, we provide the LLM with docstrings of previous terrains used for training.

```
The following is an example of an initial terrain generation function.

```
(SAME AS INITIAL EXAMPLE)
```

And the following is an example of a new, improved terrain generation function that has more
     complex obstacles. You can reference the example provided, but please make your terrain
     different. This example serves to illustrate how terrains can be made more complex, but
     you should tune the difficulty according to the previous instructions.

```
import numpy as np

def set_terrain(length, width, field_resolution, difficulty):
    """Multiple sideways-facing ramps traversing a pit for the robot to climb on and jump
     across."""

    def m_to_idx(m):
        """Converts meters to quantized indices."""
        return np.round(m / field_resolution).astype(np.int16) if not (isinstance(m, list) or
     isinstance(m, tuple)) else [round(i / field_resolution) for i in m]

    height_field = np.zeros((m_to_idx(length), m_to_idx(width)))
    goals = np.zeros((8, 2))

    # Set up platform and ramp dimensions
    # We make the platform height near 0 at minimum difficulty so the quadruped can learn to
     climb up
    platform_length = 1.0 - 0.3 * difficulty
    platform_length = m_to_idx(platform_length)
    platform_width = np.random.uniform(1.0, 1.1)   # Decrease platform width
    platform_width = m_to_idx(platform_width)
    platform_height_min, platform_height_max = 0.0 + 0.2 * difficulty, 0.05 + 0.25 *
     difficulty
    ramp_height_min, ramp_height_max = 0.0 + 0.5 * difficulty, 0.05 + 0.55 * difficulty
    gap_length = 0.1 + 0.5 * difficulty   # Decrease gap length
    gap_length = m_to_idx(gap_length)

    mid_y = m_to_idx(width) // 2

    def add_platform(start_x, end_x, mid_y):
        half_width = platform_width // 2
        x1, x2 = start_x, end_x
```

```
        y1, y2 = mid_y - half_width, mid_y + half_width
        platform_height = np.random.uniform(platform_height_min, platform_height_max)
        height_field[x1:x2, y1:y2] = platform_height

    def add_ramp(start_x, end_x, mid_y, direction):
        half_width = platform_width // 2
        x1, x2 = start_x, end_x
        y1, y2 = mid_y - half_width, mid_y + half_width
        ramp_height = np.random.uniform(ramp_height_min, ramp_height_max)
        slant = np.linspace(0, ramp_height, num=y2-y1)[::direction]
        slant = slant[None, :]  # Add a dimension for broadcasting to x
        height_field[x1:x2, y1:y2] = slant

    dx_min, dx_max = -0.1, 0.1
    dx_min, dx_max = m_to_idx(dx_min), m_to_idx(dx_max)
    dy_min, dy_max = 0.0, 0.4  # Polarity of dy will alternate instead of being random
    dy_min, dy_max = m_to_idx(dy_min), m_to_idx(dy_max)

    # Set spawn area to flat ground
    spawn_length = m_to_idx(2)
    height_field[0:spawn_length, :] = 0
    # Put first goal at spawn
    goals[0] = [spawn_length - m_to_idx(0.5), mid_y]

    # Set remaining area to be a pit
    # We do this to force the robot to jump from platform to platform
    # Otherwise, the robot can just jump down and climb back up
    height_field[spawn_length:, :] = -1.0

    # Add first platform
    cur_x = spawn_length
    dx = np.random.randint(dx_min, dx_max)
    dy = np.random.randint(dy_min, dy_max)
    add_platform(cur_x, cur_x + platform_length + dx, mid_y + dy)
    goals[1] = [cur_x + (platform_length + dx) / 2, mid_y + dy]
    cur_x += platform_length + dx + gap_length

    for i in range(1, 6):  # Set up 5 ramps
        dx = np.random.randint(dx_min, dx_max)
        dy = np.random.randint(dy_min, dy_max)
        direction = (-1) ** i  # Alternate left and right ramps
        dy = dy * direction  # Alternate positive and negative y offsets

        add_ramp(cur_x, cur_x + platform_length + dx, mid_y + dy, direction)

        # Put goal in the center of the ramp
        goals[i+1] = [cur_x + (platform_length + dx) / 2, mid_y + dy]

        # Add gap
        cur_x += platform_length + dx + gap_length

    # Add final goal behind the last ramp, fill in the remaining gap
    goals[-1] = [cur_x + m_to_idx(0.5), mid_y]
    height_field[cur_x:, :] = 0

    return height_field, goals
```

Listing 4: Evolution Example. The initial example is exactly the initial example above, and we provide an example of evolving it; in our example, we replace some flat platforms with slanted ramps.

```
Here's an alternative terrain generation function that focuses on creating a series of ramps
    for the quadruped robot to navigate through. This course will test the robot's ability to
    handle inclined surfaces and maintain balance.

'''python
import numpy as np
import random

def set_terrain(length, width, field_resolution, difficulty):
    """Ramps of varying angles for the robot to climb and descend."""

    def m_to_idx(m):
        """Converts meters to quantized indices."""
        return np.round(m / field_resolution).astype(np.int16) if not (isinstance(m, list) or
    isinstance(m, tuple)) else [round(i / field_resolution) for i in m]

    height_field = np.zeros((m_to_idx(length), m_to_idx(width)))
```

```
    goals = np.zeros((8, 2))

    # Initialize parameters
    ramp_length = 1.0 + difficulty * 2.0  # Make ramps longer and steeper with higher
     difficulty
    ramp_length = m_to_idx(ramp_length)
    ramp_height = 0.1 + difficulty * 0.4  # Ramp height varies with difficulty
    ramp_height_change = ramp_height / ramp_length  # Height change per quantized step
    ramp_width = 1.0  # Set ramp width to 1 meter
    ramp_width = m_to_idx(ramp_width)

    mid_y = m_to_idx(width) // 2

    def add_ramp(start_x, end_x, mid_y, upward=True):
        """Add a ramp either upward or downward starting from start_x."""
        for i in range(start_x, end_x):
            height = (i - start_x) * ramp_height_change if upward else (end_x - 1 - i) *
     ramp_height_change
            height_field[i, mid_y - ramp_width // 2: mid_y + ramp_width // 2] = height
            if not upward:
                height_field[i, mid_y - ramp_width // 2: mid_y + ramp_width // 2] -=
     ramp_height  # Adjust for downward ramp

    dx_min, dx_max = -0.1, 0.1
    dx_min, dx_max = m_to_idx(dx_min), m_to_idx(dx_max)
    dy_min, dy_max = -0.4, 0.4
    dy_min, dy_max = m_to_idx(dy_min), m_to_idx(dy_max)

    # Set flat ground at the start
    spawn_length = m_to_idx(2)
    height_field[0:spawn_length, :] = 0
    goals[0] = [spawn_length - m_to_idx(0.5), mid_y]  # First goal at spawn

    # Initial height is 0
    cur_x = spawn_length
    cur_upward = True  # Start with an upward ramp

    for i in range(7):  # Set up 7 ramps
        dx = np.random.randint(dx_min, dx_max)
        dy = np.random.randint(dy_min, dy_max)
        end_x = cur_x + ramp_length + dx

        add_ramp(cur_x, end_x, mid_y + dy, upward=cur_upward)

        # Place goal in the center of the ramp
        goals[i+1] = [(cur_x + end_x) / 2, mid_y + dy]

        # Switch direction (up or down)
        cur_upward = not cur_upward
        cur_x = end_x

    # Final goal at the end of the last ramp
    goals[-1] = [cur_x + m_to_idx(0.5), mid_y]
    height_field[cur_x:, :] = 0

    return height_field, goals
```

Listing 5: Example LLM Output. This output generates multiple ramps in a configuration that's distinctly different from any previously-seen terrains.

## A.3  Generated Terrain Examples

In Figure 6, we visualize some of the terrains generated by Eurekaverse. We see that the complexity and difficulty generally increases each iteration, and the quadruped learns to pass through progressively more difficult obstacle courses.

## A.4  Environment Check and Automatic Fixing

For every LLM-generated terrain function, we first check that it is executable and feasible before using it for training. We check the former by running the training script and the latter with simple heuristics:

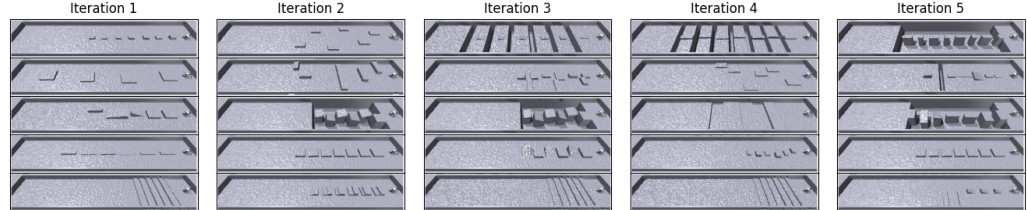

Figure 6: Visualizing a subset of our generated terrains, collected across 5 iterations within one run of `Eurekaverse`.

whether the maximum height is below 3 meters and whether the maximum height difference between goals is below 0.8 meters (double the Unitree Go1's standing height).

In our experiments, over 50% of terrains generated by GPT-4o pass the environment check. Common errors include mis-scaling the height values to incorrect units, out-of-bounds index access, and referencing missing helper functions or variables.

To save token usage and query costs, we also automatically fix invalid terrains: moving out-of-bounds goals, setting the quadruped spawn area to flat ground, and expanding obstacles that are too skinny. Note that these fixes can be easily replaced with assertions and additional LLM queries.

## A.5   Simulation Setup

**Environment Specification.** We adopt the simulation framework from Cheng et al. [21], which constructs an obstacle course terrain by quantizing the ground plane into a 2D grid and specifying the height at each index in the grid. Along with the terrain heights, the course includes 8 goal positions indicating both progression and heading the policy should follow. This specification is completely defined via a Python function, which takes a difficulty parameter as input and outputs a 2D array for terrain heights and a list of 8 $(x, y)$ coordinates for the goal positions. In `Eurekaverse`, the LLM is instructed to output precisely this function, which defines terrains $\theta_j$ modulated by the difficulty argument.

**Policy Training.** We train the policy on multiple terrains in parallel; specifically, one round of training consists of 10 terrain classes, each with 10 difficulties. Policies are initialized at low difficulties, and if they reach $g_{promote}$ goals, they are promoted to the next difficulty level; conversely, if they fail to reach $g_{demote}$ goals, they are demoted to the previous difficulty. Otherwise, they stay at the current difficulty with probability $p_{stay}$ or go to a random previous difficulty with $1 - p_{stay}$.

Following Cheng et al. [21], we first train a teacher policy using privileged scandot sensing. After fully training a privileged teacher, we then distill a student policy that takes in depth frames from a front-facing camera. Apart from the scandot or depth, the policy also receives proprioception and heading and speed commands. We train the policy with PPO [57] with heading and velocity tracking rewards, regularizations like action rate and torque, and penalties for foot placements near terrain edges. We use this same reward across all methods and ablations.

**Sim-To-Real.** To facilitate sim-to-real transfer, as in Cheng et al. [21], we use regularized online adaptation (ROA) [64], which trains an adaptation module that estimates environment properties from observation history. We also domain randomize over physical properties like friction, mass, and motor strength. During distillation, we introduce an action delay and depth sensing delay of 1 simulation step (0.02s), and we update the depth frame only once every 5 steps (10Hz). Finally, we introduce noise in the depth input, adding Gaussian noise to its true value and randomly setting some pixels to 0.

**Simulation Benchmark.** In Figure 7, we render each of the 20 obstacles in our parkour benchmark. Each of these obstacles is instantiated with 10 difficulties during evaluation, and these renders capture the obstacles at medium difficulty.

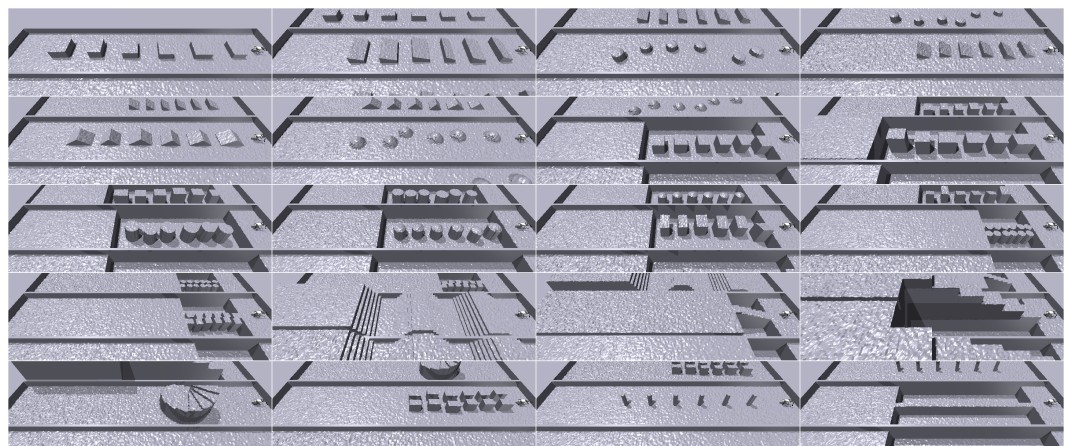

Figure 7: Visualizing our simulated parkour benchmark, roughly organized by skill: climbing boxes, walking on slopes, jumping, traversing stepping stones, climbing stairs, navigating narrow hallways, moving zig-zag through agility poles, and balancing.

## A.6 Environment Resampling

Previously, `Eurekaverse` also included a resampling mechanism: each evolution iteration, we selected high quality terrains from prior iterations for training. Specifically, terrains were chosen with probability proportional to learning progress [65–67], the difference in the policy's performance on that terrain before and after training on it. In our previous results, this increased the number of evaluation goals reached.

Additional experiments revealed that this mechanism is not necessary if we tune the inner difficulty progression (described in 5.1). Specifically, we adjusted it to diversify the parkour courses experienced across parallel agents during training, setting $g_{promote} = 8 * 0.8, g_{demote} = 8 * 0.4, p_{stay} = 0.75$. In doing so, the policy is much less likely to fall into the local minima of an individual parkour course, increasing its robustness to malformed or difficult terrain. After increasing diversity, we find that this change negates the benefit derived from environment resampling, whose effect was to avoid local minima by improving the quality of terrains used for training.

Since we observe that a careful configuration of the inner difficulty progression offsets the benefit of environment resampling, we opt to remove the latter from our method. This change streamlines our method while achieving higher evaluation results.

## A.7 Deployment Details

We deploy on the Unitree Go1, a quadrupedal robot with 12 degrees of freedom. When standing, the robot is 64.5 cm long, 28 cm wide, and 40 cm tall. We use the 3D camera mount introduced by Zhuang et al. [20] to attach an Intel RealSense D435 camera onto the Go1's head. Our vision policy runs onboard the Nvidia Jetson Xavier NX, with the depth encoder running asynchronously at 10 Hz and the policy at 50 Hz. Before depth inputs are passed through the depth encoder, we crop the left and right edges to remove dead pixels; we then apply hole-filling and temporal filters and down-sample the resolution from 270x480 to 60x90.

## A.8 Ablations

In Figure 8, we plot the performance of ablations from Figure 4 (right). All ablations plateau or degrade except for Eurekaverse.

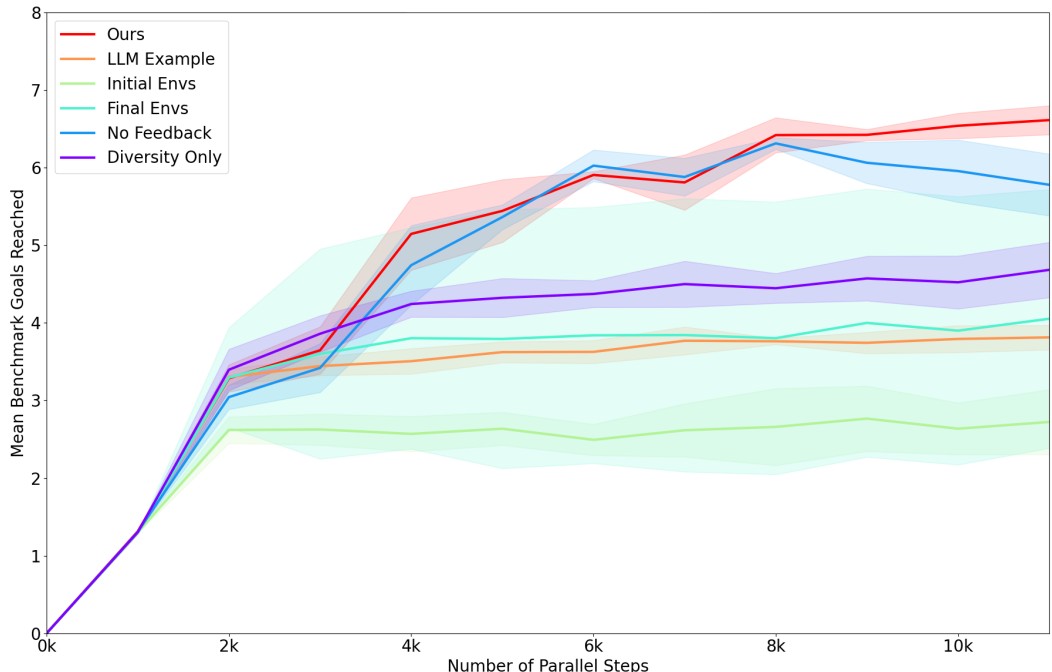

Figure 8: Comparing sim benchmark performance across training steps for Eurekaverse and ablations.

