# OpenReview forum: "Environment Curriculum Generation via Large Language Models"
_robot-learning.org/CoRL/2024/Conference — CoRL 2024_

### Official Review · Reviewer_LPUg · 2024-07-13
**Review for "Environment Curriculum Generation via Large Language Models"**

**Originality:** 3
**Technical Quality:** 5
**Clarity Of Presentation:** 5
**Potential Impact:** 3
**Recommendation:** 4
**Confidence:** 4

**Review:**

*Quality:* think it is an excellent paper that is relevant to a major bottleneck in robotics -- creating an effective curriculum for robotic applications. Besides the questions in other sections, I'm very sure that the quality of this paper is at least worth publishing if not more.

*Clarity:* the paper is very well-written. Besides a few technical issues I have not found a major fault in it.

Strengths:
1. The problem being solved here is fundamental in many RL + robotic tasks today where the agent is supposed to handle many "skills" at the same time. The solution presented here, seems applicable to many such cases.
2. The algorithm is described well, and the experiments convinced me that all proposed features were required to make it work.
3. This works well on a real system!

Weaknesses:
1. One major contribution of this paper is to alleviate "manual" environment coding. However, I would like the authors to describe all the places where human prior knowledge (such as prompt engineering) was required to reach these impressive results. I think it is required in order to understand how much this algorithm is applicable out-of-the-box for other tasks.

**Quality Of The Limitations Section:**

3

**Questions For Rebuttal:**

Besides addressing the weakness above I have the following questions:
1. I would like to understand how much of the contribution of this work is algorithmic novelty (which I suspect it mostly is!) and how much is prompt engineering? what effort did you make to make sure the baseline had the best prompt engineering available to it as well (and not one that is tailored to your algorithm)? it would be helpful to describe this process. In particular, I would like to understand the gap between the value of using a non-iterative solution compared to using your algorithm. If you think the iteration-1 and human-design reflect that, I guess I just need for you to describe why they had the best prompts they could utilize.

2. I am missing a baseline that is like iteration-1, but where the tasks are not IID during generation. I.e., every tasks is conditioned on being different from all previous tasks. This would help show that diversity is not enough, but a full curriculum where the curriculum takes into the account how the agent progresses is required. In my opinion, *initial envs* is not enough because there is no dependence between envs, and *final env* doesn't capture all the diversity the RL agent has seen (since it was initialized using past environments that might be gone by this time).

3. Compute: please discuss how much more compute you require compared to other baselines. If you require orders of magnitude more compute, discuss why running the baselines for longer will not improve their results (and of course if it will improve, show it!).

4. *Real world results:* how does the sim2real gap manifest here? If I understood correctly $\theta$, only takes care of obstacle sizes, not things like weight, friction, engines effective torque etc'. How come it is taken care of here? Is it part of the prompting or something similar? is the simulator very accurate?


Minor technical issues:
1. Line 95, the image of T should probably be a simplex over states to account for stochastic environments.
2. Line 100, mention that $\Pi$ is the space of policies.
3. Line 187: "we first pre-train a simple walking policy with 1000 parallel training steps."  What do you mean by parallel here? There is only one task -- walking.
4. Line 233: "Eurekaverse nearly matches the oracle" where can I see that. If I understand correctly figure 5 shows that there is yet a big gap in the harder tasks between oracle and iteration 5.

**Robotics Focus:**

4

**Summary Of Paper:**

The paper proposes a co-evolution algorithm that utilizes LLMs to create an automatic curriculum of environments for an RL agent to train on.

**Summary Of Recommendation:**

The clarity of presentation, quality of experiments, and the impressive results (besides my questions above), assure me that this is an excellent paper that is of high relevance to CoRL.

---

### Official Review · Reviewer_XMA4 · 2024-07-14
**very interesting work on designing environment curriculum, with limitations**

**Originality:** 4
**Technical Quality:** 3
**Clarity Of Presentation:** 3
**Potential Impact:** 3
**Recommendation:** 3
**Confidence:** 4

**Review:**

Strengths:
1. Novel approach using LLMs for unsupervised environment design in robotics, addressing the challenge of creating effective training curricula.

2. Validation on a challenging real-world task (quadrupedal parkour), with demonstration of sim-to-real transfer. Outperforms manually designed curricula, showing potential for automating complex environment design.

3. Iterative co-evolution of environments and policies enables continuous learning improvement.

Weaknesses:
1. Limited discussion on the capabilities and limitations of LLMs in generating complex, physically valid environments. It's unclear if state-of-the-art coding LLMs can effectively evolve such environments.

2. Surprising effectiveness of random sampling raises questions about the necessity of the proposed approach. A simpler solution of generating diverse environments and ordering them by difficulty might be sufficient.

3. Lack of clear explanation for emergent capabilities observed in real-world trials, such as robustness to unseen obstacles. For instance, in the staircase trials, the policy is robust to seeing chairs and railings on either side, despite never seeing similar features in simulation. What enabled this?

**Quality Of The Limitations Section:**

1

**Questions For Rebuttal:**

1. How do you ensure that the environments generated by the LLM are physically valid and increasingly complex? What are the limitations of current LLMs in this context?

2. Given the effectiveness of random sampling, how does Eurekaverse compare to simpler methods of generating diverse environments and ordering them by difficulty?

3. Can you provide more insight into the emergent capabilities observed in real-world trials? What aspects of the training process or environment design might contribute to this robustness?

=== post rebuttal
The authors have made a good effort to address my comments. I am happy with 2. and 3. I am more positive now. However, there is no "accept" rating. I do not think the paper crossed the threshold to be a strong accept.

**Robotics Focus:**

4

**Summary Of Paper:**

This paper introduces Eurekaverse, an unsupervised environment design algorithm that uses large language models (LLMs) to generate progressively more challenging and diverse environments for training robotic skills. The key idea is to leverage LLMs' code generation capabilities to create varied obstacle courses for quadrupedal parkour learning. The authors validate Eurekaverse on a quadruped robot traversing various obstacle courses, demonstrating effective sim-to-real transfer. The automatically generated curriculum enables gradual learning of complex parkour skills in simulation and successful transfer to real-world scenarios, outperforming manually designed training courses.

**Summary Of Recommendation:**

I recommend a weak accept for this paper. The proposed Eurekaverse algorithm presents a novel and potentially impactful approach to unsupervised environment design using LLMs. The validation on quadrupedal parkour and demonstration of sim-to-real transfer are significant strengths. However, the paper would benefit from addressing the limitations mentioned, particularly regarding the capabilities of LLMs in generating complex environments, the comparison to simpler methods, and the explanation of emergent capabilities. With these improvements, the paper could make a valuable contribution to the field of robotic learning and environment design.

---

### Official Review · Reviewer_VvEd · 2024-07-21

**Originality:** 3
**Technical Quality:** 5
**Clarity Of Presentation:** 5
**Potential Impact:** 3
**Recommendation:** 4
**Confidence:** 4

**Review:**

The paper is well-written and has a good flow. The authors motivate the problem well and cover the relevant related work that has been performed in this domain in the recent past. The authors combine ideas such as co-evolution (used in other problem domains like robot design), and using the baked-in knowledge and reasoning capabilities of LLMs and apply it to a previously unexplored setting of environment/terrain curriculum generation. The ablations make sense and serve to highlight the importance of the different components used in the pipeline.

Strengths:
- The authors have done a great job with extensive real-world evaluation of their method.
- The implementation details, including the algorithm pseudocode, prompts used, examples, and progress reweighing function serve to increase the clarity and reproducibility of the proposed method.

Weaknesses:
- Given that the proposed method uses an evolutionary algorithm style population-based method, it can prove to be quite expensive, depending on the value of N and J chosen.

**Quality Of The Limitations Section:**

2

**Questions For Rebuttal:**

- Given that this is an automated method, what is the wall-clock time taken from start to finish? What computing setup is used to train the policies?
- What is the cost in USD for obtaining the final policy that can be deployed on the real robot?

**Robotics Focus:**

4

**Summary Of Paper:**

Curriculum learning is an effective technique for training an agent to perform increasingly difficult and complex tasks. However, in most cases, the curriculum is designed by domain experts. In the current paper, the authors propose leveraging LLMs for generating the environment curriculum, which is over different terrains. The method starts by initializing an i.i.d. set of environments and then training a population of agents on subsets of all the environments. Each agent is then evaluated on the entire set of environments, and the best performing agent is retained. This agent serves as the initialization for the population in the next iteration. The next step in the method involves querying the LLM to output a new set of environments that can be used to improve the policy’s performance further. The method then alternates between these two steps for several iterations to finally obtain the best performing policy. The authors also evaluate their method on the real robot.

**Summary Of Recommendation:**

While the idea of using an LLM to act as a “domain expert” in itself is not novel, the authors use it in a novel setting, backed by a compelling and thorough set of real world testing experiments, demonstrating the efficacy of their proposed method.

---

### Author Rebuttal · Authors · 2024-08-07

Dear AC and reviewers,

We thank you for your time and effort in reviewing and providing detailed suggestions towards improving our work\! We have updated our manuscript (attached in this message, edits are highlighted in red) with additional details and experiments. We feel that these new additions have greatly strengthened our paper:

1. Explanation of the compute and API cost used per run.
2. Clarification on our **Random Resampling** ablation and our method’s “inner” curriculum during training, which orders terrains based on difficulty.
3. Explanation of our policy’s failure recovery and robustness to distractors in real-world trials.
4. A baseline, **Diversity Only**, that modifies **Initial Envs** to increase environment diversity by conditioning LLM queries on past generated environments.
5. A baseline, **No Feedback**, that generates environments and orders them by difficulty via an LLM. Specifically, this runs our method without including policy feedback in the evolution prompt, instead asking the LLM to make the environments harder each iteration.
6. Various fixes to wording and details.

Below, we have also responded to each reviewer’s questions and comments. We thank all reviewers again for their time and effort helping us improve our paper\! Please let us know if we can provide additional clarifications or explanations.

Best,
Authors

---

### Decision · Program_Chairs · 2024-09-04

**Decision:**

Accept

**Comment:**

Strengths:
1. The paper introduces a novel approach using  LLMs for generating environment curricula, leveraging their code generation capabilities for creating progressively more challenging environments.
2. Demonstrates effective sim-to-real transfer and validates the method on real-world tasks, specifically quadrupedal parkour.
3. The paper is well-written with a good flow, providing detailed implementation, pseudocode, examples, and a progress reweighing function, enhancing clarity and reproducibility.
4. Addresses a fundamental problem in robotics and RL tasks, making the solution potentially applicable to various scenarios requiring multiple skills.
5. The experiments are thorough, showing the necessity of different components in the pipeline and highlighting the efficacy of the proposed method.

Weaknesses:
1. The population-based evolutionary algorithm can be computationally expensive, and the paper lacks a discussion on the wall-clock time and computing setup required.
2.  The surprising effectiveness of random sampling raises questions about the necessity of the proposed method compared to simpler solutions that generate diverse environments ordered by difficulty.
3. The paper does not clearly explain the emergent capabilities observed in real-world trials, such as robustness to unseen obstacles.
4. This work is missing comparisons with baselines that could use simpler non-iterative solutions or diverse environment generation conditioned on being different from previous tasks.